# Participating in core outcome set development via Delphi surveys: qualitative interviews provide pointers to inform guidance

Alice M Biggane [1,2] Paula R Williamson,[1] Philippe Ravaud,[2,3,4] Bridget Young[5]

¹Biostatistics, University of Liverpool, Liverpool, UK
²CRESS, INRA, INSERM, Université de Paris, Paris, France
³Epidemiology, Columbia University, New York City, New York, USA
⁴Centre d'Épidémiologie Clinique, AP-HP (Assistance Publique des Hôpitaux de Paris) Hôpital Hôtel Dieu, Paris, France
⁵Health Services Research, University of Liverpool, Liverpool, UK

**Correspondence to**
Dr Alice M Biggane;
abiggane@liverpool.ac.uk

## ABSTRACT

**Objectives** To explore participants' views of Delphi surveys in core outcome set (COS) development.
**Study design and setting** Patients and health professionals (n=24) from seven recently concluded COS studies that had involved a Delphi survey took part in semistructured qualitative interviews (telephone and email exchange). Interviews explored participants' understanding of COS and their experiences of the Delphi survey. Analysis was thematic.
**Results** Several interviewees had previously participated in two or more COS or Delphi surveys. Those with multiple experiences of participation generally understood the purpose of COS and were satisfied with the Delphi survey. However, some interviewees who were first-time participants struggled to understand the purpose of COS and aspects of the Delphi survey, which limited their contribution and satisfaction with the study. Interviewees also differed in how they interpreted and subsequently used the written documentation provided to COS participants. Some interviewees wanted guidance regarding whose perspective to take into account when scoring outcomes and on how to use the scoring system. Interviewees reported being motivated to take part by the international and expert consensus aspects of the Delphi survey. A few interviewees reported experiencing either positive or negative emotional impacts arising from when they reviewed outcomes and stakeholder feedback.
**Conclusion** This study identifies important information that should be communicated to COS Delphi study participants. It also indicates the importance of communicating about COS Delphi studies in ways that are accessible and salient to participants, to enhance their experience of participation and make the process more meaningful for all.

## INTRODUCTION

Inconsistency in outcomes measured in clinical trials is a major concern across a multitude of health conditions, limiting the synthesis of available evidence and ability to reach reliable conclusions.[1 2]

Core outcome sets (COS) are one potential solution to this problem. A COS is a minimum set of agreed standardised outcomes which should be measured and reported in all trials

**Strengths and limitations of this study**

► This is the first study to explore, in depth, the experiences of patients and health professionals who took part in core outcome set (COS) development via the Delphi survey.
► A strength of this study is that we were able to ask interviewees specific, tailored questions thus exploring their personal perspectives and insights of COS Delphi study participation.
► This study sampled an international selection of patients and health professionals.
► This study sampled from COS Delphi studies in a range of health conditions.
► Limitations include the retrospective nature of the interview.

in a specific condition as a minimum.[3] Three important stakeholder groups in the development of COS for trials are health professionals, patients and those who will use the COS in research, such as clinical trialists or industry.[4]

Several methods are used to include stakeholders as participants in COS development, including interviews, focus groups, nominal group technique and Delphi surveys. Delphi surveys, used singularly or in combination with other methods, are the most popular method of facilitating participation.[5] These involve iterative rounds of questionnaires listing outcomes and asking participants to score the importance of each outcome. Scores are subsequently summarised across the various stakeholder groups and fed back to participants in the following round. This allows participants to consider the views of others before rescoring each item. Furthermore, participants' views are anonymised which minimises the influence of power differentials between different stakeholders that can be problematic with direct communication between participants.[6 7] The creation,

**BMJ**

administration and analysis of Delphi surveys are relatively inexpensive. The availability of online Delphi survey platforms allows large samples and facilitates international development of COS, thus, ensuring they are relevant globally.

However, Delphi surveys have been described as potentially intimidating for some patient participants[7] and COS developers have acknowledged a need for guidance on conducting Delphi surveys and the consensus meetings which typically follow them.[8] While recent surveys of COS participants indicate that their experiences of Delphi surveys have been generally favourable,[9 10] no research has explored in depth the perspectives of patients and health professionals on participating in COS Delphi surveys. We therefore explored their opinions and experiences of participation to identify ways to enhance Delphi surveys for future participants in COS studies.

## METHODS
### Research design
In the current study, Exploring Participant Input in Core Outcome Set Development, taking a broadly pragmatic approach, we used semistructured qualitative interviews to explore patients' and health professionals' experiences of participating in COS Delphi surveys.

### Sampling strategies and recruitment
We used the responses of COS developers to a previous survey[5] to inform purposeful sampling of host COS studies from which to recruit interviewees. This survey was informed by searches of the Core Outcome Measures in Effectiveness Trials (COMET) Initiative database. COMET has created and maintains a publicly accessible database (www.comet-initiative.org) of planned, ongoing and completed COS projects and is updated annually with published studies that have been identified through a systematic review. The survey was sent to all COS developers who had published or registered a study with COMET since 2013. Host studies were eligible if they had involved a Delphi survey, had patient participants, included participants from more than one country and had concluded no more than 6 months prior to the interview. COS developers of each host study distributed a recruitment advert (online supplementary file 1) to all stakeholders who registered for the first round of the Delphi survey. The advert invited interested individuals to contact AMB who provided a participant information sheet. For each host COS, we aimed to interview up to two patients and two health professionals. Interviewees were sent a thank you card and £15 (or currency equivalent) shopping voucher as an acknowledgement.

### Data collection
Interviewees were geographically dispersed so were interviewed via telephone or email exchange. The data were collected between October 2017 and June 2018. At the time of interview, interviewees were between 7 months and 6 weeks from having participated in the final round of the host COS Delphi. All telephone interviews were semistructured and used a topic guideline which allowed for a conversational approach to be adopted to explore issues that we anticipated to be important, while enabling interviewees to raise areas that were important to them. COS developers and public contributors with experience of COS development informed the initial development of the topic guide (online supplementary file 2), as did previous qualitative research.[11] Ongoing data analysis informed the further iterative development of the topic guide. Furthermore, the interviewer, AMB, tailored questions for each interviewee by reviewing available information on the host study prior to every interview. This information included, for example: participant information materials such as guidance sheets and videos, the number of rounds, scoring systems used, numbers of domains and outcomes scored and examples of outcomes scored. For one host study, a screenshot of the Delphi survey was supplied by the developers which AMB then used as a memory aid with interviewees from that COS Delphi study. Email interviews followed a similar format asking a range of open-ended questions across topics, if necessary the interviewer, AMB followed up on responses with additional open-ended questions to further explore the interviewees' answers and comments. All interviewees gave informed consent. The first two audio-recorded interviews were transcribed verbatim by AMB, and the remainder were transcribed verbatim by a University of Liverpool approved transcription agency into Microsoft Word. Transcripts were checked and anonymised before being analysed. The data are currently held in password-encrypted files on The University of Liverpool's secure server. AMB, who was a PhD student supervised by PRW and BY, conducted all interviews in English. Before starting data collection, she received training in qualitative methods.

### Data analysis
Data analysis drew on Braun and Clarke's six-phase thematic approach.[12] Analysis was initially deductive following the topic guides but became more inductive as the analysis progressed[12] and ranged from line-by-line coding, to considering whole transcripts. AMB initially read the transcripts and reflective fieldnotes that she had made immediately after each interview to inform her interpretations. A codebook was developed for the content using open coding. By grouping the codes together, recurring patterns and themes were identified and organised into categories.[12] AMB led the analysis, which she periodically discussed with BY and PRW, who each read a sample of the transcripts and reviewed reports of the developing analysis. All three agreed that data saturation (the point at which new data cease to contribute to the analysis) had been reached after 24 interviews. Microsoft Word was used to facilitate coding and analysis.[13] While accepting that quality procedures cannot promise

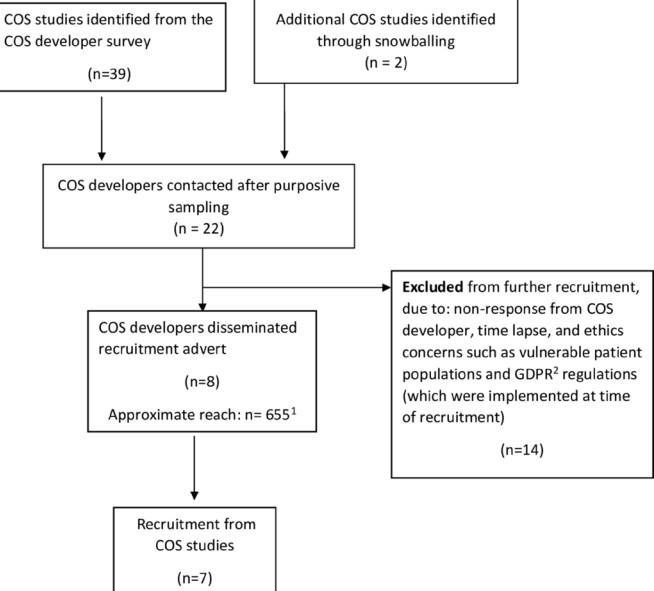

**Figure 1** Sampling of core outcome set (COS) studies that fit our sampling framework. [1]Reach of two COS studies is unknown, approximate relates to the other six COS studies. [2]General Data Protection Regulation (GDPR) is a European Union (EU) law regulation regarding data protection and privacy for all individuals within the EU and the European Economic Area.[15]

quality,[14] the reporting of this study was informed by relevant guidance.[15]

### Patient and public involvement statement

Patients and the public were involved in developing and reviewing the topic guide, recruitment advert and participant information sheets used in this study.

### Definitions

We use the term 'patient' to refer to patients, carers, service users and people from organisations who represent these groups. We use 'health professional' to refer to clinicians and pharmacists. Interview excerpts shown below were selected to demonstrate the findings and our interpretations. Health professionals are indicated by 'HP' and patients by 'P', the COS in which they took part is indicated by 'COS' and a number, for example, HP1COS1; "[.......]" indicates text removed for succinctness.

## RESULTS

### COS study sampling and interviewee characteristics

We initially identified 39 potential host COS studies via the survey[5] (figure 1). Two further ongoing COS studies were brought to our attention by COS developers, which were not in the COMET database at the time of the survey, but were subsequently added. We contacted the developers of 20 of these COS studies in batches to inform purposive sampling to achieve maximum variation. Of these 20, we excluded 14 studies from further consideration (figure 1). We distributed our recruitment advert, via the COS developers, to the participants in the remaining six

COS studies, plus the two further studies brought to our attention, giving eight unique online COS studies. Of these, we recruited participants from seven COS studies. These studies covered: geriatrics (COS1), dermatology (COS2), other (COS3), cancer (COS4), paediatrics (COS5), gynaecology and obstetrics (COS6) and otorhinolaryngology (COS7), and all aimed to recruit international participants.

They varied in terms of the number of outcomes to be scored, the number of rounds, scoring system and in the ways feedback was presented to Delphi survey participants.

Following distribution of our advert, 40 participants from the seven COS studies contacted us. We did not take forward interviews with 11 of these (6 health professionals, 2 patients and 3 unknown status) as interview quotas for their COS study had been reached. Of the 29 participants invited for interview, 24 participated. Of the remaining five, two patients withdrew as they were unable to recall any details of their COS study while two patients and a health professional did not respond after the initial contact.

Table 1 summarises the demographic characteristics of the 24 interviewees, 2 interviews were completed by email exchange, and the remainder were telephone interviews. Twelve (50%) were resident in the UK, four in Ireland, three in Canada and one from each of Australia, Italy, Singapore, Spain and the Netherlands. Twenty-two interviewees described themselves as having professional occupations, two patient interviewees were retired and did not disclose their most recent occupation. Ten interviewees (three patients and seven health professionals) had previous experience of COS, Delphi surveys or both. One of the three patients with previous experience was also the patient research partner (involved in the design and conduct) of the COS development about which they were interviewed.

### Findings from interviews

For most interviewees, taking part in an online Delphi survey several months ago had not been a particularly salient or memorable event. Therefore, some interviewees, particularly patients, at times struggled to recall details of the host COS and so the interviewer had to provide them with brief prompts or reminders throughout the interviews. For example, P9COS5 had '*signed up to a lot of studies*' during the same time period, and asked the interviewer to remind her of what the study was about. On explaining the topic of the Delphi survey and giving some reminders of the process such as the number of rounds and the process of reviewing and scoring outcomes, P9COS5 commented that she could recall filling out only one round of the Delphi survey. Thus, her interview is in relation to that only.

While all participants in each of the seven COS studies had access to resources such as information sheets (and to online videos for two of COS studies), which explained the purpose and format of the study, interviewees differed

**Table 1** Interviewee demographic characteristics

| Identifier | Gender | Age range (years) | Country | Prior participatory experience | |
| --- | --- | --- | --- | --- | --- |
| | | | | COS development | Delphi survey |
| P1COS1 | Male | 65–74 | UK | No | No |
| P2COS1 | Female | ≥75 | UK | No | No |
| P3COS2 | Female | 35–44 | UK | No | No |
| P4COS3 | Female | Undisclosed | Canada | Yes | Yes |
| P5COS2 | Male | 45–54 | UK | No | No |
| P6COS3 | Female | 55–64 | Canada | Yes | Yes |
| P7COS4 | Female | 55–64 | UK | No | No |
| P8COS4 | Female | 55–64 | Netherlands | No | No |
| P9COS5 | Female | 35–44 | Ireland | No | No |
| P10COS6 | Female | 45–54 | Ireland | Yes* | Yes |
| P11COS7 | Male | 55–64 | UK | No | No |
| P12COS7 | Female | 65–74 | UK | No | No |
| P13COS2 | Female | 55–64 | UK | No | No |
| HP1COS1 | Female | 45–54 | Canada | No | Yes |
| HP3COS4 | Male | 45–54 | Spain | Yes | Yes |
| HP4COS2 | Female | 35–44 | Singapore | Yes | Yes |
| HP5COS4 | Male | 35–44 | UK | Yes | Yes |
| HP6COS5 | Female | 55–64 | UK | No† | No |
| HP7COS5 | Female | 25–34 | Ireland | No‡ | No |
| HP8COS5 | Female | 35–44 | UK | No† | No |
| HP9COS5 | Female | 65–74 | Ireland | Yes | Yes |
| HP10COS6 | Female | 35–44 | Italy | No | No |
| HP11COS6 | Male | 65–74 | UK | Yes | Yes |
| HP12COS6 | Female | 55–64 | Australia | Yes | Yes |

*Interviewee was also the patient research partner of the core outcome set (COS) study they were interviewed in relation to.
†Two health professionals stated awareness/knowledge of COS and Delphi survey but had not participated previously.
‡One health professional was involved in an earlier phase of the COS study for which they participated in the Delphi survey.

in how accurately and fully they understood the purpose of COS and the process of the Delphi survey.

### Synthesis and interpretation

In what follows, we present five thematic findings from our interviews as follows: (1) how previous experience helped interviewees understand COS Delphi studies, (2) the differences in how participants understand the processes and purposes of Delphi surveys, (3) the question of who is being represented in the COS Delphi studies, (4) the motivational and emotional aspects of COS Delphi participation and (5) how the scoring system used in Delphi surveys are understood by participants

### Previous experience helped interviewees understand COS Delphi studies

As indicated in table 1, several interviewees had previous experience of COS and Delphi surveys. In comparison to those without such experience, these interviewees generally showed a better understanding of the purpose of COS and indicated greater satisfaction with the Delphi survey. HPs with previous experience (n=7) praised COS for their importance and usefulness in research, and the Delphi survey method for its simplicity. HP5COS4 said "that's the beauty of it, it is just not a difficult, all the hard work is done by the people that analyse the data. It is just like answering a customer service survey from Sky isn't it? Click next, next, next you just do it don't you, but I would put more effort to this than I would do a customer survey from Sky because it is more important to me".

HPs without previous experience talked about having about read up to COS and Delphi surveys or of seeking advice from colleagues and peers to enhance their understanding of the study and prepare for their participation. For example, HP7COS5 took part in an earlier event for the same COS study at which the developers had been present; "it made me think more fully about the bigger picture of research going forward and how these processes like the Delphi survey feed into that" and that otherwise she "would have approached it in a less informed way."

Three patient interviewees also spoke about the impact of their previous experiences in COS Delphi studies. Over the course of these studies, they described their experience evolving from one of confusion during their first study to one of enjoying the process and better understanding the purpose of COS with each subsequent study "once you get the hang of it, I really enjoy doing them because I like where it takes you" (P6COS3). P10COS6 spoke of not having a "bull's notion what is going on" in earlier studies with regard to both the purpose and method of COS development and had "to do a lot of online research myself to learn", despite receiving information sheets for each study. Reflecting on this evolving experience of COS and Delphi surveys during her interview, she suggested that providing participants with a visual synopsis of the purpose of COS and Delphi survey method from the outset of a study would be helpful: "I would have assimilated the message much quicker".

Patient interviewees (n=9) with no previous experience varied in their understanding of the purpose of the Delphi survey. The comments of some showed that they understood the Delphi survey's purpose was to reach consensus on which core outcomes to include. For example, P7COS4 explained the study was: "looking at how people felt with their recovery […] what they went through and what they were left with and how important those were to the person involved". In contrast, others such as P8COS4 described the Delphi survey's purpose more vaguely as to gather a "broad base of information on how many different people experience the treatment". Moreover, she did not talk about the process in terms of prioritising the outcomes listed or reaching consensus among stakeholders. P1COS1 was confused about whether his study was complete or if he should expect further rounds of the survey: "I don't even know that you could say a line had been drawn under it". P11COS7 reflected on whether he "could have done more to understand how the process worked earlier on. Particularly with the […] expert involvement, I now understand so next time I shall be even better at it" and suggested "a practice run" would have been useful before entering the actual study. In a few cases participants indicated that their lack of understanding had influenced their overall experience of participation, "I think one of my real concerns is that I didn't really contribute anything to the research because I really wasn't sure what I was doing" (P2COS1).

### Helping participants understand the purpose and process of Delphi surveys: one size does not fit all

The findings indicate that interviewees had different needs for support to aid their understanding of the purpose and process of COS Delphi surveys. P3CSO2 and P5COS2 were two first-time patient participants. They both received the same study documentation and said they reviewed it. However, their accounts indicated that they differed in their understanding of the documentation, and these differences influenced their contributions to and experiences of the study.

P5COS2 thought the study documentation he received was "appropriate", elaborating "I have worked in the past in IT, in pharmaceuticals, in politics[…]so I am quite happy to see text that is fairly technical in nature or fairly clinical in nature and you know that is something I find easy enough to get to grips with". He thought that the study "was a very constructive thing to do. And I could see personally, something like that being done prior to any clinical trial, so that the end points of the clinical trial […] look at, you know how beneficial say a product is from the patient's perspective".

In contrast, P3COS2 who worked in marketing commented that she "didn't understand the terminology" in the documents and as a result described being "switched off from the process element […] psychologically I was just focussed on taking part and having my say". She wondered if the study and its data would get "stored away somewhere in a filing cabinet and forgotten about [….] I think what was lacking in the communication is how this is going to actually practically inform future research. And maybe that is my lack of understanding of how these sort of surveys work, and how these outcome surveys work, I don't really get, how that will translate into future treatments". In response to P3COS2's comment, the interviewer explained that COS were used as minimum sets of outcomes in clinical trials so that evidence can be compared across studies and inform decision-making regarding treatments. The interviewer added that the Delphi survey was a method to develop the COS by seeking consensus among relevant experts including patients. In response, P3 recalled that she had received information to that effect in the study documentation before adding "I really wish that had been captured in the communication a bit more clearly […] maybe I'd have done things differently".

### Representation in the Delphi survey: who and when

Both HP and patient interviewees raised the issue of "who they should be representing?" when completing the Delphi survey. They questioned whether they should try to think or imagine what outcomes fellow patients or HPs would likely prioritise when scoring the outcomes study, or whether they should focus only on their own opinions and priorities. None reported receiving guidance on this.

P5COS2 thought "it can only be a genuine result if everybody says what they personally feel" and "trying to guess [how others feel]" would defeat the objective. This contrasts with P7COS4, a female who described trying to answer the outcomes section of that was applicable to males only: "I just thought well if I was in that situation I will answer it as if I was that person maybe you know. […] Yes maybe I shouldn't have done that".

In COS3, both patient interviewees were also advocates in a relevant patient organisation, and both had previous experience of COS Delphi studies. P6 described how she "learned very early on" to answer from her own perspective. Conversely, P4 drew on her knowledge of the perspectives of other patients from discussions she had

had through her work with the patient organisation "I do try to work in their concerns and the issues that they have". She added that COS developers should consider how the different phases in a patient's journey and their life could affect the way they scored outcomes: "my priorities are different now, than they were when I was diagnosed over 30 years ago […] you know different things would have affected me. […] over the years with the chronic disease you learn to live with it and adapt to it, so […] yes I think that can affect your responses too".

HPs touched on similar issues regarding who to represent when scoring outcomes, although compared with patients, this was less prominent in their accounts. HP1COS1, was an academic, a service provider and a policy-maker. Referring to both her experiences as a professional and her personal opinions, she explained that she drew on "a bit of both" when scoring outcomes. Similarly, HP11COS6, an academic and service provider, explained "it was a mixture of, of relating it to myself and relating it to patients. But I was, even when I was relating it to myself I was relating it to me thinking of myself as a patient or the father of a patient or something like that".

### Motivational and emotional aspects of participation
A few patients and HPs talked about the motivational and emotional aspects of their participation.

Health professionals praised the Delphi survey method of COS development for its consensual and collaborative approach, and cited the opportunity to learn from international colleagues as one of the motivations for participating. They also spoke of their belief in the importance of COS in their field and their desire to contribute.

Patients described being 'happy' that they could contribute their experiential knowledge and have input in research studies relevant to them. Some saw the COS study as one of the few research projects relevant to their condition and this was a motivating factor in their participation. P8COS4 talked about how her illness was 'rare' and how information and research on the illness was limited "so it was great for us (other patients) and for me specifically you know to fill in something that was specifically to do with my (illness)", she further elaborated that the COS study "made us feel someone was listening or someone was going to help us". P3COS2 talked about how she felt happy to be included in research relevant to her, as she was outside the age range that was typical for patients with the health condition concerned. Similarly, P5COS2 "thought it was quite exciting the fact that they would ask regular kind of sufferers of particular problems what do you think should be included in a trial. What outcomes do you think are important and everything and getting feedback from people outside the scientific community. I thought was quite cool and as somebody who suffers from various medical conditions the ability for me to give my input on what I think is important to a patient".

P6CO3 had participated in multiple COS Delphi studies. She described her enthusiasm for the Delphi survey as a motivation to participate: "every time I do them, I enjoy them more I really, really like the process" and her willingness to participate in studies that used the resulting COS: "you might have a preconceived notion of what something should be, or perspective on what something should be, or what the final product should look like, and it takes you in a different direction and if you just kind of you know let go and let it take you where it takes you through the questions and the feedback and everything I think it is a really interesting way of coming up with a list and I think it is a really true list".

Two patients and one health professional indicated that reviewing the list of outcomes had affected them emotionally. Speaking of when she reviewed the scores provided by fellow participants in the second round of P8COS4 commented that she had: "changed some of my answers on the second round, when I was thinking about having a possible (intervention removed) then I was like oh, I wouldn't want that at all [… ] I was sort of realising that I was grateful for where I was basically". HP7COS5 said that when reviewing the fellow participants' feedback "there were definitely moments of almost insecurity I suppose because you are aware, […] you are in amongst a group of other people who are very familiar with this field and experts […]". She described initially feeling uncertain about her answers: "it is ok to obviously be encouraged to check back on yourself and to be really thoughtful when you are kind of giving those sorts of answers [….] so I think there was a little bit of both an awareness of needing to stay objective but there was certainly a more subjective, emotive aspect to seeing how other people were answering".

P2COS1 spoke of how reviewing the outcomes as part of the COS study had made her aware of outcomes that she had not previously realised were associated with her condition and treatment: "A lot of the outcomes I would never have thought of those as outcomes from the sort of medication I am on if you see what I mean". She described how this had affected her: "I am seriously worried about that. […] I was given no indication [by healthcare provider] […] that I need to be careful".

### Scoring system
The scoring systems in the seven host COS studies used either a 9- (n=6) or 5- (n=1) point Likert scale. In five of the COS that used a 9-point Likert, scores were further differentiated as: 1–3 'Not important' (n=4) or 'Limited importance' (n=1), 4–6 'Important but not critical', 7–9 'Critical'. In the sixth, the anchor descriptions were 'not at all important'[1] and 'extremely important'.[9] In the COS that used a 5-point Likert scale, participants were asked to rate their level of agreement on a series of statements regarding potential outcomes, with scores labelled: 1 Strongly disagree, 2 Disagree, 3 Ambivalent, 4 Agree and 5 Strongly agree.

Several interviewees did not comment on the scoring system during their interview. Those who did comment varied from praising or indicating satisfaction with the

scoring system, to wanting a system with fewer categories and further guidance on how to apply the scale, although the majority of interviewees were positive about the scales used in COS Delphi studies that they had taken part in. Those who expressed satisfaction with the 9-point scales, indicated that they were familiar with using these: "I am usually happy with Likert scales so, fine" (HP12COS6), while another interviewee summed up her experience of the scales as "not a big deal" (P4COS3).

Interviewees who took part in a COS that used a 9-point scale and liked it praised the wide range of options and the three distinct bands as helpful. For example, HP9COS5 commented "I liked the way they set it out in that they were, you know while it was 9 it was important, not so important and least important so that even within those categories one could actually subdivide them, and I actually think I liked that. Sometimes you know you are asked you know, should something be important, and there are kind of gradations within importance, and so I think that for me I liked that subdivision. It gave me a little bit more flexibility". P7COS4 noted "grading it you know, systematically up from 1 to 9 so yes that was useful because it give you, although a lot of my scores were up on the higher range there were a couple of lower ones so I think the having 1 to 9 was a good idea".

Other interviewees had a preference for fewer categories. Speaking of the 9-point scale in her study, P2COS1 commented "I really don't think a score from 1 to 10 is realistic. […] maybe if you are a very skilled researcher yourself you might be able to deal in that level of gradation but I don't think the vast majority of us can. I think, you know, a 5 point rating scale is the most that most of us could do. You know with any degree of accuracy". Similarly, also speaking of the 9-point scale HP8COS5 said "what is the difference between a 6 and 7, you know what I mean if it is just sort of all in the middle of the road […] so whether or not it could have been less numbers to help make a more definitive answer". However, like other interviewees who had a preference for a scale with fewer categories she acknowledged "there might be reasonings behind why you have got 0–9 and that type of thing". While some interviewees found the three bands on the 9-point scale helpful, responses from some health professionals and patients indicated that further guidance and support are needed to help them use the 9-point scale. Similarly P11COS7, a first time patient participant, raised the difficulties he experienced in "connecting physical sensations with a numerical value" when relating his physical symptoms to scoring outcomes. He added that this "produces a certain anxiety between whether you pick 5, 6 or 7".

HP6COS5 was the only interviewee who compared the scoring system to other methods of prioritisation when she flagged her overall preference for a numerical scale when scoring a long list of items in comparison to ranking them "if I had been given the list and said you know rate these 1 to 20 it would have been harder to do".

## DISCUSSION
### Summary of findings
We found that while some interviewees understood the purpose of COS and the Delphi survey, others struggled to understand the purpose and aspects of the Delphi survey method which in turn influenced their contribution and experience of the study. The accounts of the interviewees indicate that COS participants would benefit from further guidance and support.

Interviewees could be broadly separated into two categories: those with and without previous experience of COS development and/or Delphi surveys. The accounts of those with previous experience, both health professionals and patients, showed they had a good understanding of the purpose of COS and were satisfied with the Delphi survey as a method of participation. Health professionals without previous experience reported engaging with relevant literature and colleagues prior to and during participation, thus enhancing their understanding and experience. In contrast, the accounts of patients without previous experience indicated considerable variation with some showing good understanding, while others understood little of the study and its purpose. Aspects that the latter group struggled with included understanding that the Delphi survey aimed to achieve consensus among stakeholders, applying the scoring system and knowing whose views to represent when participating. This limited their engagement and interpretation of the documentation they had received from COS developers, and their input and experience of COS development.

The importance of representing of all relevant stakeholder groups including patients in COS development[4 7] is increasingly recognised, as it is in wider health research.[16–18] There is also growing appreciation of the importance of supporting their participation in ways that are meaningful, thus avoiding tokenism and enhancing the credibility and validity of the resulting research.[19 20] However, our findings suggest that not all the interviewees thought their participation in COS development was meaningful, as the purpose and process of the study were communicated in ways that were not accessible for them. Theory surrounding health literacy describes its role in patient empowerment and advocates for information to be made accessible to all patients in appropriate formats[21–24] This is particularly important for patient participants in COS development, most of whom will not have taken part in this type of research previously nor have access to the literature or colleagues to illuminate the process. A few patient interviewees in this study indicated that they saw understanding COS Delphi studies as their personal responsibility or felt uncomfortable with their limited of understanding. However, when asking patients to participate in COS studies, developers are inviting them to the world of research[7]; thus, it is the responsibility of the COS development community to ensure the guidance and support are in place to allow meaningful participation. There has been a rapid expansion in the number of COS being developed, with an associated rapid increase in the

number including patients in Delphi surveys. Our findings indicate that this expansion has perhaps outpaced the development of relevant guidance for Delphi studies to enable meaningful participation for all.

> **Box 1  Summary of the pointers and recommendations core outcome set (COS) developers should consider when designing and conducting their COS Delphi studies**
>
> **Pointers**
> ► COS developers should consider the most appropriate medium(s) to communicate their COS Delphi studies information and guidance
> *Points to consider: language used, target audience, health condition*
> ► COS developers need to ensure that the scoring system used is explained in ways that participants can understand.
> ► COS developers should explain to participants whose perspectives they should consider when scoring in different rounds.
> ► COS developers should explain to participants that in the first round of the Delphi survey they should score outcomes according to their own individual perspective.
> *Proxies: in the first round, COS developers should ask proxies to score according to what they anticipate is the perspective of the patient and not from their own perspective as a carer*
> ► COS developers should ask participants in second or subsequent rounds to reflect on the scores of other participants, while also being clear that participants do not have to change their own scores.
> *Proxies: should follow the same advice as other participants in second or subsequent rounds*
> ► COS developers can encourage participants to score outcomes they have no experience of to date, but may experience in the future, although an 'unable to score' option or equivalent should also be provided for each outcome.
> ► COS developers should consider the potential influence of their COS Delphi on participants and take appropriate steps to minimise negative effects.
> ► By understanding what motivates participants into COS Delphi studies, COS developers can devise appropriate recruitment and retention strategies.

This study points to specific areas where further guidance and support is required to communicate the purpose of COS and the process of the Delphi survey, which we summarise as pointers for COS developers to consider in box 1. This complements the findings of two recent surveys of COS Delphi study participants which indicated that they benefit from repeated guidance on principles of COS development during the rounds, that reminders about these principles were acceptable,[10] and that recruitment and retention of participants are more likely with personalised communication.[9] To date, the most common way of providing participant information regarding a research project is via written documentation. Much research has indicated poor health literacy is prevalent[25–28]; thus, the importance of ensuring plain language communication cannot be underestimated. However, this study's findings suggest that plain language communication, and further consideration of how to explain the purpose of COS in ways that are relatable and salient to patients are required. This explanation and delivery could make use of visual, written and auditory methods,

such as analogies, infographics, visual metaphors, digital stories and other narrative forms. The most appropriate method or combination of methods is likely to depend on the population and health condition to which the COS will be relevant. The use of visual resources has been documented in other healthcare areas such as health promotion,[29] patient education[30] and nursing training.[31] In COS, development demonstration videos of the Delphi survey enhanced participant retention to the study.[9] The COMET website provides resources to help developers facilitate participation, including documents explaining COS in plain English and an animation video (http://www.comet-initiative.org/resources/PlainLanguageSummary), coproduced with members of the public.

This study also indicates areas in which further research and direction would be useful. The issues raised by interviewees regarding how to apply the scoring system point to the need for better communication. The 9-point Likert scoring system where items are graded in accordance to their level of importance is a common method, recommended by the Grading of Recommendations Assessment, Development and Evaluation Working Group.[32] There are statistical considerations in support of using a longer scale including the ability to calculate variance in scores. Thus, it is important that participants in COS Delphi studies have the information and support they need to apply this system. Involving patients and members of the public as active research partners would provide a patient perspective on the suitability of different aspects of the COS study from design to conclusion, including helping with the development of appropriate documentation, resources and support.[7 9]

Interviewees also raised the issue of whose perspective to take into account when scoring outcomes. Pending further research, we would recommend that in the first round of the Delphi survey COS developers ask participants to score according to their own individual perspective, not score according to the perspective of others. In the second or subsequent rounds, participants should be asked to reflect on the scores of other participants, while being clear that they do not have to change their own scores. Having reflected participants should be asked to score according to their current view of what a COS in that specified health condition should include.[33] Participants can be encouraged to score outcomes they have no experience of to date, but may experience in the future, although an 'unable to score' option or equivalent should also be provided for each outcome. A key exception to participants scoring from their own individual perspective is when carers act as proxy respondents in COS studies. In health research on certain patient populations, there is often no alternative to using proxies,[34 35] yet there is evidence of discrepancies in how proxies prioritise outcomes compared with patients themselves.[36 37] During the first round of COS Delphi studies, proxies should score according to what they anticipate is the perspective of the patient and not from their own perspective as a carer, and follow the same advice as other participants

in subsequent rounds. Thus, COS developers should consider which proxies can provide a valid opinion on the anticipated perspective of the patient and how best to support this type of participation.

Some interviewees described the motivation and emotions associated with their participation. Understanding that participants are motivated to engage in COS development out of desire to contribute to the research topic and satisfaction with the Delphi survey's collaborative and international approach will be useful to COS developers when advertising and recruiting participants to their study. The emotional impact of participation requires consideration from developers and researchers when designing and conducting their COS studies to optimise the experience of participants and minimise any negative impacts on them.

## Strengths and weaknesses of the study

This study has provided insights into COS development via Delphi surveys from the perspective of participants. As previously noted, participation in the COS Delphi studies was not a particularly salient event for interviewees; however, during their interviews, they were provided with tailored prompts and reminders as needed.

This study only describes the experiences of participants who agreed to be interviewed, recruited from seven COS studies and limited to English speakers. Those interviewed, including patients, mostly described themselves as having 'professional backgrounds'. Thus, while saturation was reached within our sample, we note that interviewees' experiences and perspectives may not but typical of the wider patient population. However, by purposively sampling across a range of COS studies, we anticipate that our findings will be broadly transferable to other COS studies. Moreover, our interviewees were international, reflecting the increasing international development of COS.

## CONCLUSION

This study's findings contribute to the growing evidence base on participation in COS development. The identification of areas where participants need enhanced guidance and support will be useful to future COS developers when planning their studies, enabling them to recruit and support participants towards a meaningful and positive experience of COS Delphi studies.

**Acknowledgements** The authors would like to thank the COS developers and the interviewees for their time and input. They would like to acknowledge the Methods in Research on Research network for their support and guidance, in particular Van Thu Nguyen, Ketevan Glonti and Melissa Sharp. They are also grateful to Camila Olarte Parra and Els Goetghebeur for their thoughtful comments and suggestions on the final draft of the manuscript.

**Contributors** All authors have read and approved the final version of this manuscript. AMB was the lead researcher on this project and was responsible for the preparation and drafting of the protocol, data collection and analysis and writing of this manuscript. PRW is a coinvestigator and contributed to project conception, design, protocol writing, analysis, writing and proofreading of this manuscript. PR is a coinvestigator and contributed to the methodology used in creating the sampling framework. BY is principal investigator on this project and is responsible for its design, protocol writing, analysis, writing and proofreading of this manuscript.

**Funding** This project is a part of a Methods in Research on Research (MiRoR) funded PhD, which is being undertaken by AB. MiRoR has received funding from the European Union's Horizon 2020 research and innovation programme under the Marie Sklodowska-Curie grant agreement No 676207.

**Competing interests** PRW chairs the Management Group of the COMET Initiative. BY is a member of the COMET People and Public Participation, Involvement and Engagement Working Group.

**Patient consent for publication** Not required.

**Ethics approval** Ethical approval was granted from Health and Life Sciences Committee on Research Ethics (human participants, tissues and databases) at The University of Liverpool, on 22/06/2017 (reference 1969). All interviewees were provided with full written information prior to interview commencement. All interviewees gave audio recorded or written informed consent prior to commencing the interview and were free to end the interview at any time without providing a reason.

**Provenance and peer review** Not commissioned; externally peer reviewed.

**Data availability statement** No data are available.

**ORCID iD**

Alice M Biggane http://orcid.org/0000-0001-6568-6486

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
