## [Reviewer comments · BMJ Open]

ARTICLE DETAILS

TITLE (PROVISIONAL)	Participating in core outcome set development via Delphi surveys: Qualitative interviews provide pointers to inform guidance
AUTHORS	Biggane, Alice M; Williamson, Paula; Ravaud, Philippe; Young, Bridget

VERSION 1 – REVIEW

REVIEWER	Kathryn Fackrell University of Nottingham, United Kingdom I have developed a COS and conducted a COS Delphi study
REVIEW RETURNED	03-Sep-2019

GENERAL COMMENTS	This study is well-written and given the raise in Delphi surveys for COS development involving patients is very timely and important. By interviewing patients and professionals, this study has provided in-depth information on the experiences of participating in COS Delphi studies and the problems that can arise through these studies which are not always clear when designing and conducting the Delphi studies. This study provides important information about the process and how to improve the experiences of participants in these remote studies such as these where research support is not always immediately on hand and there is a reliance on information that is provided. Therefore, this study is important. I have few comments to make as the paper efficiently addresses the description of the methods. Reporting guidelines were referenced in the ethics section, but some important information that is recommended is missing, such as where is the data being held? Who transcribed the interviews into which software? What software was used for the data analysis? When was the data collected? The dates of the first interview and last. What is the time frame participants were having to recall? (smallest – longest). Also it would improve clarity if the subsections in the reporting guidelines were reflected in the manuscript. For example, separating data collection and data analysis, using sampling strategies instead of participants as the participant characteristics are reported in the results. It would also be more useful for the use of these guidelines to be reported at the beginning of the methods section not at the end. The flow of the data collection and analysis section feels interrupted and the information being reported, for example on the development of the topic guide, can scattered throughout the section. This makes it hard to follow the data collection methods. To improve clarity and understanding, it would be helpful to reported all the information for the development of the topic guide
--

and general information on the interviews together before you report the specific information for the email interviews and telephone interviews (i.e. audio recorded).
For example, initially it is not clear that the development of the topic guide was an iterative process as this is reported until later on the section. It would be more informative to reader if this information is upfront with the fact that the initial topic guide was informed by experiences of COS developers and public contributors and later updated based on interim analysis.
Also the following sentence (page 4, Lines 27-29) “AMB tailored questions for each interviewee by reviewing available information on the host study before the interviews.” currently it reads as if this was only done for email interviews because of where it is reported in the paragraph. I think what you have done here is really interesting and should be highlighted that you did this for all interviews and how this helped. I would also suggest giving an example of the type of information that you might use from the host study for those not as familiar with Delphi and COS.

Minor

Methods

Page 4, Lines 9-11. In this section refer to purposeful sampling when reporting the eligibility criteria so it is clearer in results to what the criteria is for your purposeful sample.

Page 4, Line 20: “They were topic-guided and semi-structured, using a conversational approach to explore issues that we anticipated to be important, while enabling interviewees to raise areas that were important to them”. This sentence is hard to follow. I am unsure who or what you are referring to by “they”. Is this telephone interviews or email emails, or both? Do you mean that each interview was semi-structured and used a topic guide which allowed for a conversational approach to be adopted to explore...

Page 4, Line 30-31. “Due to the international focus of this study, interviews were conducted via telephone or email exchange.” Suggest removing as already mentioned above that interviews were conducted via telephone or email due to be geographically dispersed.

Page 4, Lines 36-41, Page 5, Lines 1-4. In the abstract it states that thematic analysis was conducted, but this is not clear in the methods section as there is no mention of thematic analysis being undertaken.

Page 7, Lines 4-6. “We initially identified 39 potential host COS studies via the survey [5] (Supplementary File 3. Figure 1). Two further ongoing COS studies were brought to our attention by COS developers, which were not in the COMET database at the time of the survey, but were subsequently added.” For readers that are not familiar with COMET and the database, the sentence above is confusing. Please clarify what COMET is an abbreviation of and the purpose of the database. I suggest making it clear that the COS developers were identified via a survey that was conducted using the studies listed in the COMET database.

Results

Page 10, Lines 1-14. I am not sure about the subsection title “findings from the interviews”. The section appears to report recall problems and differences in the COS that the participants would be recalling. I suggest changing the heading for this section to reflect the characteristics being reported here.

Page 10, Line 15. It would be informative to direct the reader on the number of themes identified and their names before going into detail on each theme. I suggest using the subheading “synthesis

	and interpretation” for this thematic results section to make it clear that these results are now being reported. Page 10, Line 25. Type in text, replace the second ‘about’ with ‘to’ Line 8-9. “Patients described taking part out of gratitude for a study in which they could contribute their lived experience.” I am unsure of what is meant by ‘taking part out of gratitude’. Patients participated because they were happy to be asked to contribute their experiences to research? To share the knowledge? To make changes? Discussion The discussion nicely brings together the findings and makes some recommendations for future COS Delphi studies. It would be good to have a summary of the recommendations (provided in table?) related to each problem identified for researchers to use when developing protocols for their COS Delphi survey.
--	--

REVIEWER	Susan J Bartlett McGill University, Canada
REVIEW RETURNED	11-Sep-2019

GENERAL COMMENTS	This is a very well written manuscript that addresses an important knowledge gap – namely the experiences of individuals participating in Delphi exercises to identify core outcome sets. (I was particularly intrigued with the results given we are in the midst of planning another Delphi to do this with a broad group of stakeholders.) Overall, I was impressed with the writing, the level of detail, the rigor of the data collection, and the thoughtful interpretation. The international focus of the manuscript and scope of diseases included are also important strengths. The only substantive comment I have is with respect to the participants. It is interesting to note that 22 of the interviewees described themselves as professionals. This seems to be a common occurrence where increasingly patient research partners are also health care providers or researchers themselves. Further, 10 of the participants had previous experience participating in Delphi’s to develop core outcome sets, and 1 of the 3 patients was also the patient research partner (involved in the design and conduct of the COS development. It is indeed an important finding that despite most having an advanced education, many struggled to understand what exactly was asked of them and what would be done with the results. I would encourage the authors to add some discussion about how the characteristics of the sample may have influenced the results obtained. This is particularly important given that professionals are likely to differ from the general population of patients in important ways. Perhaps a “sensitivity” analysis might be considered where highly experienced or highly educated individuals are excluded.
---

VERSION 1 – AUTHOR RESPONSE

Reviewer(s)' Comments to Author:

Reviewer: 1

Reviewer Name: Kathryn Fackrell

Institution and Country: University of Nottingham, United Kingdom

Please state any competing interests or state 'None declared': I have developed a COS and conducted a COS Delphi study

Please leave your comments for the authors below

This study is well-written and given the raise in Delphi surveys for COS development involving patients is very timely and important. By interviewing patients and professionals, this study has provided in-depth information on the experiences of participating in COS Delphi studies and the problems that can arise through these studies which are not always clear when designing and conducting the Delphi studies. This study provides important information about the process and how to improve the experiences of participants in these remote studies such as these where research support is not always immediately on hand and there is a reliance on information that is provided. Therefore, this study is important. I have few comments to make as the paper efficiently addresses the description of the methods.

Thank you kindly for taking the time to review our paper. We are grateful for your suggestions and have answered each comment individually below.

1. Reporting guidelines were referenced in the ethics section, but some important information that is recommended is missing, such as where is the data being held? Who transcribed the interviews into which software? What software was used for the data analysis? When was the data collected? The dates of the first interview and last. What is the time frame participants were having to recall? (smallest – longest).

Thank you for your comment. We used the Standards for Reporting Qualitative Research (SRQR)* checklist by O'Brien et al for this paper. It was included as a supplementary file in which all the suggested items were accounted for. However, based on your suggestion above we agree these items are also useful and have included them in appropriate places throughout the methods section as follows:

- The data is currently held in password encrypted files on The University of Liverpool's secure server.
- The first two audio recorded interviews were transcribed verbatim by AMB, the remainder were transcribed by a University of Liverpool approved transcription agency into Microsoft Word. Transcripts were checked and anonymised before being analysed.
- Microsoft Word was used to facilitate coding and analysis (REF)
- The data were collected between October 2017 and June 2018.
- At the time of interview, interviewees were between seven months and six weeks from having participated in the final round of the host COS Delphi

2. Also it would improve clarity if the subsections in the reporting guidelines were reflected in the manuscript. For example, separating data collection and data analysis, using sampling strategies instead of participants as the participant characteristics are reported in the results. It would also be more useful for the use of these guidelines to be reported at the beginning of the methods section not at the end.

Thank you for your suggestion. We have moved the guideline reference to the beginning of the section as you suggest. We have also changed the heading from "Participants and recruitment" to "sampling strategies and recruitment." We have also divided "data collection" and "data analysis" into two separate sections.

3. The flow of the data collection and analysis section feels interrupted and the information being reported, for example on the development of the topic guide, can be scattered throughout the section. This makes it hard to follow the data collection methods. To improve clarity and understanding, it would be helpful to report all the information for the development of the topic guide and general information on the interviews together before you report the specific information for the email interviews and telephone interviews (i.e. audio recorded).

For example, initially it is not clear that the development of the topic guide was an iterative process as this is reported until later on the section. It would be more informative to reader if this information is upfront with the fact that the initial topic guide was informed by experiences of COS developers and public contributors and later updated based on interim analysis.

Thank you for your suggestion. Based on this and the comment above we have edited both sections so they read as follows:

“Data collection:

Interviewees were geographically dispersed so were interviewed via telephone or email exchange. The data were collected between October 2017 and June 2018. At the time of interview, interviewees were between seven months and six weeks from having participated in the final round of the host COS Delphi. All telephone interviews were semi-structured and used a topic guideline which allowed for a conversational approach to be adopted to explore issues that we anticipated to be important, while enabling interviewees to raise areas that were important to them. COS developers and public contributors with experience of COS development informed the initial development of the topic guide (Supplementary File 2), as did previous qualitative research [11]. Ongoing data analysis informed the further iterative development of the topic guide. Furthermore, the interviewer, AMB, tailored questions for each interviewee by reviewing available information on the host study prior to every interview. This information included, for example: participant information materials such as guidance sheets and videos, the number of rounds, scoring systems used, numbers of domains and outcomes scored and examples of outcomes scored. For one host study a screenshot of the Delphi survey was supplied by the developers which AMB then used as a memory aid with interviewees from that COS Delphi study. Email interviews followed a similar format asking a range of open-ended questions across topics. If necessary, AMB followed up on responses with additional open-ended questions to further explore or clarify the interviewees’ answers and comments. All interviewees gave informed consent. The first two audio-recorded interviews were transcribed verbatim by AMB, the remainder were transcribed verbatim by a University of Liverpool approved transcription agency into Microsoft Word. Transcripts were checked and anonymised before being analysed. The data is currently held in password encrypted files on The University of Liverpool’s secure server. AMB, who was a PhD student supervised by PRW and BY, conducted all interviews in English. Before starting data collection, she received training in qualitative methods.

Data analysis:

Data analysis drew on Braun and Clarke’s six phase thematic approach (Ref). Analysis was initially deductive following the topic guides but became more inductive as the analysis progressed [12] and ranged from line-by-line coding, to considering whole transcripts. AMB initially read the transcripts and the reflective fieldnotes that she had made immediately after each interview to inform her interpretations. A codebook was developed for the content using open coding. By grouping the codes together, recurring patterns and themes were identified and organised into categories [12]. AMB led the analysis, which she periodically discussed with BY and PRW, who each read a sample of the transcripts and reviewed reports of the developing analysis. All three agreed that data saturation (the point at which new data cease to contribute to the analysis) had been reached after twenty-four interviews. . Microsoft Word was used to facilitate coding and analysis (REF)

Pg. 4 Line 24-42, Pg. 5 Line 1-18

4. Also the following sentence (page 4, Lines 27-29) “AMB tailored questions for each interviewee by reviewing available information on the host study before the interviews.” currently it reads as if this was only done for email interviews because of where it is reported in the paragraph. I think what you have done here is really interesting and should be highlighted that you did this for all interviews and how this helped. I would also suggest giving an example of the type of information that you might use from the host study for those not as familiar with Delphi and COS.

Thank you for your suggestion. We have now added the following sentence to elaborate what we did:

“Furthermore, AMB, tailored questions for each interviewee by reviewing available information on the host Delphi study prior to every interview. This information included, for example: participant information materials such as guidance sheets and videos, the number of rounds, scoring systems used, numbers of domains and outcomes scored and examples of outcomes scored. For one host study a screenshot of the Delphi survey was supplied by the developers which AMB then used as a memory aid with interviewees from that COS Delphi study.”

Pg. 4 Line 33-39

Minor
Methods

5. Page 4, Lines 9-11. In this section refer to purposeful sampling when reporting the eligibility criteria so it is clearer in results to what the criteria is for your purposeful sample.

Thank you for your suggestion. This now reads as:

“We used the responses of COS developers to a previous survey [5] to inform purposeful sampling of host COS studies from which to recruit interviewees.

Pg. 4 Line 8

6. Page 4, Line 20: “They were topic-guided and semi-structured, using a conversational approach to explore issues that we anticipated to be important, while enabling interviewees to raise areas that were important to them”. This sentence is hard to follow. I am unsure who or what you are referring to by “they”. Is this telephone interviews or email emails, or both? Do you mean that each interview was semi-structured and used a topic guide which allowed for a conversational approach to be adopted to explore...

Thank you for your comment. We have now edited the relevant section to clarify what we meant. It reads as:

“All telephone interviews were semi-structured and used a topic guideline which allowed for a conversational approach to be adopted to explore issues that we anticipated to be important, while enabling interviewees to raise areas that were important to them. COS developers and public contributors with experience of COS development informed the initial development of the topic guide (Supplementary File 2), as did previous qualitative research [11]. Ongoing data analysis informed the further iterative development of the topic guide. Furthermore, the interviewer, AMB, tailored questions for each interviewee by reviewing available information on the host study prior to every interview. This information included, for example: participant information materials such as guidance sheets and videos, the number of rounds, scoring systems used, numbers of domains and outcomes scored and examples of outcomes scored. For one host study a screenshot of the Delphi survey was supplied by the developers which AMB then used as a memory aid with interviewees from that COS Delphi study. Email interviews followed a similar format asking a range of open-ended questions across topics, if necessary the interviewer, AMB followed up on responses with additional open-ended questions to further explore the interviewees’ answers and comments. All interviewees gave informed consent.”

Pg.4 Line 27-42

7. Page 4, Line 30-31. “Due to the international focus of this study, interviews were conducted via telephone or email exchange.” Suggest removing as already mentioned above that interviews were conducted via telephone or email due to be geographically dispersed.

We have removed this line as per your suggestion.

8. Page 4, Lines 36-41, Page 5, Lines 1-4. In the abstract it states that thematic analysis was conducted, but this is not clear in the methods section as there is no mention of thematic analysis being undertaken.

Thank you for bringing this to our attention. We have added the following sentence to the beginning of the relevant section which reads as follows and includes an appropriate reference from Braun and Clarke.

“Data analysis drew on Braun and Clarke’s six phase thematic approach (Ref).”
Pg.5 Line 8

9. Page 7, Lines 4-6. “We initially identified 39 potential host COS studies via the survey [5] (Supplementary File 3. Figure 1). Two further ongoing COS studies were brought to our attention by COS developers, which were not in the COMET database at the time of the survey, but were subsequently added.” For readers that are not familiar with COMET and the database, the sentence above is confusing. Please clarify what COMET is an abbreviation of and the purpose of the database. I suggest making it clear that the COS developers were identified via a survey that was conducted using the studies listed in the COMET database.

Thank you for raising this important point. We have added the following explanation to section 2.2, as it is where we first introduce the survey that was used to inform our sampling. The additional text reads as:

“This survey was informed by searches of the COMET (Core Outcome Measures in Effectiveness Trials) Initiative database. COMET has created and maintains a publicly accessible database (www.comet-initiative.org) of planned, ongoing and completed COS projects and is updated annually with published studies that have been identified through a systematic review. The survey was sent to all COS developers who had published or registered a study with COMET since 2013.

Pg. 4 Line 9-14

Results

10. Page 10, Lines 1-14. I am not sure about the subsection title “findings from the interviews”. The section appears to report recall problems and differences in the COS that the participants would be recalling. I suggest changing the heading for this section to reflect the characteristics being reported here.

Thank you for this suggestion. While we appreciate your comment on this section, we would prefer to include this section as findings rather than with the characteristics of the interviewees. We believe the insight that COS Delphi studies are not particularly salient or memorable events is an important finding in its own right, and that it also provides context for understanding the themes in the following subsections.

11. Page 10, Line 15. It would be informative to direct the reader on the number of themes identified and their names before going into detail on each theme. I suggest using the subheading “synthesis and interpretation” for this thematic results section to make it clear that these results are now being reported.

Thank you for your suggested edits. We have added the following to indicate the number of themes that are reported in the subsequent sections, under the suggested heading.

“In what follows we present five thematic findings from our interviews as follows: i) how previous experience helped interviewees understand COS Delphi studies, ii) the differences in how participants understand the processes and purposes of Delphi surveys, iii) the question of who is being represented in the COS Delphi studies, iv) the motivational and emotional aspects of COS Delphi participation and v) how the scoring system used in Delphi surveys are understood by participants.”

Pg. 10 Line 16-21

12. Page 10, Line 25. Type in text, replace the second ‘about’ with ‘to’

Thank you. We have edited this as suggested.

13. Line 8-9. “Patients described taking part out of gratitude for a study in which they could contribute their lived experience.” I am unsure of what is meant by ‘taking part out of gratitude’. Patients participated because they were happy to be asked to contribute their experiences to research? To share the knowledge? To make changes?

Thank you for raising this point. We have now edited this section to clarify our point. Our edits read as:

“Patients described being “happy” that they could contribute their experiential knowledge and have input in research studies relevant to them. Some saw the COS study as one of the few research projects relevant to their condition and this was a motivating factor in their participation.”

Pg.14 Line 8-11, Section 3.2.4

Discussion

14. The discussion nicely brings together the findings and makes some recommendations for future COS Delphi studies. It would be good to have a summary of the recommendations (provided in table?) related to each problem identified for researchers to use when developing protocols for their COS Delphi survey.

Thank you for this very helpful suggestion. We have now added the following pointers (in keeping with the title of the manuscript) table to the Discussion.

Pointers

- COS developers should consider the most appropriate medium(s) to communicate their COS Delphi studies information and guidance

Points to consider: Language used, target audience, health condition

- COS developers need to ensure that the scoring system used is explained in ways that participants can understand.

- COS developers should explain to participants whose perspectives they should consider when scoring in different rounds

- COS developers should explain to participants that in the first round of the Delphi survey they should score outcomes according to their own individual perspective.

Proxies: In the first round, COS developers should ask proxies to score according to what they anticipate is the perspective of the patient and not from their own perspective as a carer

- COS developers should ask participants in second or subsequent rounds to reflect on the scores of other participants, while also being clear that participants do not have to change their own scores.

Proxies: should follow the same advice as other participants in second or subsequent rounds

- COS developers can encourage participants to score outcomes they have no experience of to date, but may experience in the future, although an “unable to score” option or equivalent should also be provided for each outcome.

- COS developers should consider the potential influence of their COS Delphi on participants and take appropriate steps to minimise negative effects.

- By understanding what motivates participants into COS Delphi studies, COS developers can devise appropriate recruitment and retention strategies

Reviewer: 2

Reviewer Name: Susan J Bartlett

Institution and Country: McGill University, Canada

Please state any competing interests or state ‘None declared’: None

Please leave your comments for the authors below

This is a very well written manuscript that addresses an important knowledge gap – namely the experiences of individuals participating in Delphi exercises to identify core outcome sets. (I was particularly intrigued with the results given we are in the midst of planning another Delphi to do this with a broad group of stakeholders.) Overall, I was impressed with the writing, the level of detail, the rigor of the data collection, and the thoughtful interpretation. The international focus of the manuscript and scope of diseases included are also important strengths.

Thank you kindly for taking the time to review our paper. We are grateful for your suggestions and have answered your comment below.

The only substantive comment I have is with respect to the participants. It is interesting to note that 22 of the interviewees described themselves as professionals. This seems to be a common occurrence where increasingly patient research partners are also health care providers or researchers themselves. Further, 10 of the participants had previous experience participating in Delphi's to develop core outcome sets, and 1 of the 3 patients was also the patient research partner (involved in the design and conduct of the COS development. It is indeed an important finding that despite most having an advanced education, many struggled to understand what exactly was asked of them and what would be done with the results.

I would encourage the authors to add some discussion about how the characteristics of the sample may have influenced the results obtained. This is particularly important given that professionals are likely to differ from the general population of patients in important ways. Perhaps a "sensitivity" analysis might be considered where highly experienced or highly educated individuals are excluded. Thank you for your suggestion. In light of your comments above we agree that this an important point to acknowledge in our paper and have edited the limitations section of our discussion to reflect this. It now reads as

"This study only describes the experiences of participants who agreed to be interviewed, recruited from seven COS studies and limited to English-speakers. Those interviewed, including patients, mostly described themselves as having "professional backgrounds". Thus, while saturation was reached within our sample we note that interviewees' experiences and perspectives may not but typical of the wider patient population. However, by purposively sampling across a range of COS studies, we anticipate that our findings will be broadly transferable to other COS studies. Moreover, our interviewees were international, reflecting the increasing international development of COS." Pg. 21 Line 9-12

VERSION 2 – REVIEW

REVIEWER	Kathryn Fackrell University of Nottingham, United Kingdom
REVIEW RETURNED	14-Oct-2019

GENERAL COMMENTS	Thank you for opportunity to review this revised manuscript, I have enjoyed reading such a well-written and clear article with really interesting results. The authors have made a number of changes that have improved the clarity of methods and results. This study will provide some key vital information for those developing COS using Delphi studies. There are no further comments to make about the paper.
--

REVIEWER	Susan J Bartlett McGill University, Canada
REVIEW RETURNED	19-Sep-2019

GENERAL COMMENTS	The authors have addressed almost all concerns. The acknowledgement of patients mostly describing themselves as having "professional backgrounds" in the discussion is important as it raises the issue of generalizability of results to non-professionally trained patients. It should also be noted in the summary of strengths and limitations below the abstract.
--